# Multi-station automatic classification of seismic signatures from the Lascar volcano database

Pablo Salazar[1,2,3], Franz Yupanqui[1,6], Claudio Meneses[4], Susana Layana[1], and Gonzalo Yáñez[5]

[1]Millennium Institute on Volcanic Risk Research - Ckelar Volcanoes, Avenida Angamos 0610, Antofagasta, Chile.
[2]Departamento de Ciencias Geológicas, Universidad Católica del Norte, Chile.
[3]Centro de Investigación para la Gestión Integrada del Riesgo de Desastres (CIGIDEN), Santiago, Chile.
[4]Departamento de Ingeniería de Sistemas y Computación, Universidad Católica del Norte, Chile.
[5]Departamento de Ingeniería Estructural y Geotécnica, Pontificia Universidad Católica de Chile, Chile.
[6]Programa de Doctorado en Ingeniería Sustentable, Universidad Católica del Norte, Chile.

**Correspondence:** Pablo Salazar (pasalaz@ucn.cl)

**Abstract.** This study was aimed to build a multi-station automatic classification system for volcanic seismic signatures such as hybrid, long period, tremor, tectonic and volcano-tectonic events. This system was based on a probabilistic model made using transfer learning, which has, as the main tool, a pre-trained convolutional network named AlexNet. We designed five experiments using different datasets with data that was real, synthetic, two different combinations of these (combined 1 and combined 2), and balanced subset without synthetic data. The experiment presented the highest scores when a process of data augmentation was introduced into processing sequence. Thus, the lack of real data in some classes (imbalance) dramatically affected the quality of the results, because the learning step (training) was over-fitted to the more numerous classes. To test the model stability with variable inputs, we implemented a k-fold cross-validation procedure. Under this approach, the results reached high predictive performance, considering that only the percentage of recognition of the tectonic events (TC) class was partially affected. The results obtained showed the performance of the probabilistic model, reaching high scores over different test datasets. The most valuable benefit of using this technique was that the use of volcano seismic signals from multiple stations provided a more generalisable model which, in near future, can be extended to multi-volcano database systems. The impact of this work is significant in the evaluation of hazard and risk by monitoring the dynamic evolution of volcanic centres, which is crucial for understanding the stages in a volcano's eruptive cycle.

## 1 Introduction

### 1.1 The problem of monitoring

The task of detecting the seismic activity of an active volcano and the subsequent characterisation (classification) of these events are, in many cases, the most time-consuming in observatories worldwide. This is because of the massive amount of data that is collected daily in a continuous record, by a single seismic network with a few stations. In this context, big data analysis tools have become an attractive option for reaching levels of processing that have never been achieved using traditional techniques.

This study proposed the use of machine learning and transfer learning techniques to automatically classify volcanic seismic events. Determining the type of volcanic event in a continuous seismic record (time series) will facilitate the construction of a model of evolution associated with the dynamics of a volcanic centre. This should create better understanding and evaluation of the hazards and risks associated with volcanic activity, improving efforts made in this matter (e.g., de Natale et al. , 2019; Magrin et al. , 2017; Rapolla et al. , 2010). The novelty of this approach is in its use of previously trained deep convolutional networks, such as AlexNet, in a scenario that considers the information recorded by a network of multiple stations. It permits variability in the input of data and improves the generalisation of the system. The generalisation of the system directly impacts the performance of the training models of pattern recognition for each class and creates the possibility of applying the generated models in a common multi-volcano database system in the near future.

## 1.2 A summary of methods used in Volcano Seismic Recognition

Among the classification techniques, methods based on a probabilistic approach and hidden Markov models (HMMs) are most relevant. The advantages of HMMs include the possibility of managing data with different durations, computational efficiency, and an elegant interpretation of results based on Bayes' theorem (Carniel, 2014). Several studies have been performed on volcanic systems using this technique with different approaches. Continuous HMMs were used for the simultaneous detection and classification of continuous volcanic responses (Beyreuther and Wassermann , 2008), whereas discrete HMMs were applied to analyse and classify events as described by Ohrnberger (2001). Other applications for HMMs were considered in the works of Bebbington (2007), who used the method to analyse a catalogue of flank eruptions recorded at Mt. Etna. Hidden semi-Markov models were applied by Beyreuther and Wasserman (2011) using time dependence to improve the performance of the method. Beyreuther et al. (2012) also introduced state clustering to improve the time discretisation in induced seismicity experiments. Contrastingly, Bicego et al. (2013) used an HMM method, based on a hybrid generative-discriminative classification paradigm, in pre-triggered signals recorded at the Galeras Volcano in Colombia.

Other classification techniques, such as artificial neural networks, provide an efficient approach for the classification of not only seismic events, but also time slices of continuous signals, such as volcanic tremors (Carniel, 2014). The multi-layer perceptron (MLP) is often used for the analysis of seismic signals recorded at volcanoes. Esposito et al. (2013) applied the MLP technique for landslide recognition, while Esposito et al. (2014) utilized MLP to estimate the possible trend of the seismicity level in Campi Flegrei (Italy). Self-organising maps (SOM), another class of artificial neural networks, have been used to analyse very long period events at the Stromboli volcano (Esposito et al. , 2008), as well as volcanic tremors at the Etna volcano (Langer et al. , 2009, 2011), Raoul Island volcano (Carniel et al. , 2013a), and Ruapehu volcano (Carniel et al. , 2013b). Furthermore, self-organising maps with time-varying structures (SOM-TVS) have been applied to volcanic signals to achieve improvements in relation to SOMs (Araujo and Rego , 2013).

Notably, the support vector machine (SVM) approach developed by Vapnik (1995), which is based on linear discrimination, should be mentioned. For a two-class problem, SVM uses linear elements for discrimination, i.e., lines, planes, or hyperplanes. Masotti et al. (2006, 2008) used this technique in analysing volcanic tremor data recorded at Mt. Etna in 2001. Langer et al. (2009) applied this approach to compare several supervised and unsupervised pattern-classification techniques. Ceamanos et

al. (2010) built a multi-SVM classifier for remote-sensing hyperspectral data. The simultaneous application of SVM and MLP was also performed by Giacco et al. (2009), who used the two methods to discriminate between explosion quakes, landslides, and tremors recorded at the Stromboli volcano.

Thus, numerous studies have been conducted to develop an automated system for the detection and classification of volcanic signals. The early systems consisted of classifiers that used data from a single station to design different approaches (Masotti et al. , 2006; Beyreuther and Wassermann , 2008; Rouland et al. , 2009; Langer et al. , 2011; Bicego et al. , 2015). However, after some years, the systems evolved into more complex algorithms that facilitated the building of models using the information from a few stations or channels (Z, E, N). Nevertheless, they did not use the data from all the possible stations in the network, instead their results were based on one station or channel that was used as a pattern (Álvarez et al. , 2012; Esposito et al. , 2013; Carniel et al. , 2013b; Cortés et al. , 2014, 2015; Curilem et al. , 2014a, b; Bicego et al. , 2015). Interestingly, the work of Curilem et al. (2016), based on station-dependent classifiers, shows the possibility to mix information from different stations to create models that enable the classification of events at different stations, despite the fact that experiments were performed with a reduced database.

## 1.3 Supervised machine learning as strategy for automatisation

The systems that allow us to build probabilistic model for an automatic classification of volcanic event are called Volcano-Seismic Recognition (VRS). The probabilistic models are built from data determined previously by an expert geophysicist. The models obtained are later used over continuous seismic records for automatic and unsupervised classification.

As previously mentioned, pattern recognition and automatic classification require the previous classification of seismic signals into different classes, making this one of the most important, but also one of the most time-consuming, tasks when accomplished daily by a human operator.

The study was aimed to present a novel approach that considered a supervised machine-learning strategy (transfer learning) using AlexNet, a previously trained deep convolutional neural network, to create a multi-station automatic classification system for volcanic seismic signatures.

Transfer learning for deep neural networks is the process of first training a base network on a source dataset and then transferring the learned features (network weights) to a second network to receive training on a target-related dataset. From a practical point of view, the reuse or transfer of information from previously learned tasks for the learning of new tasks has the potential to significantly improve the efficiency of a reinforcement learning agent.

AlexNet was the first convolutional network which used GPU to boost performance. Its architecture consists of 5 convolutional layers, 3 max-pooling layers, 2 normalization layers, 2 fully connected layers, and 1 softmax layer (Azrina et al. , 2019). Each convolutional layer consists of convolutional filters and a nonlinear activation function ReLU. The pooling layers are used to perform max pooling. Input size is fixed due to the presence of fully connected layers, and is mentioned at most of the places as 224x224x3. However, due to some padding it works out to be 227x227x3. Overall, AlexNet has 60 million parameters.

In 2012, AlexNet won the ImageNet visual object recognition challenge, i.e. the ImageNet Large Scale Visual Recognition Challenge (ILSVRC) (Krizhevsky et al. , 2012). The numbers of classes to be classified by ImageNet dataset consist of 1,000. Therefore the final fully connected layer also contains 1,000 neurons. The ReLU activation function is implements to the first seven layers respectively. A dropout ratio of 0.5 is applied to the sixth and seventh layer. The eighth layer output is finally supplied to a softmax function. Dropout is a regularization technique, being used to overcome the overfitting problem that remains a challenge in a deep neural network. Thus, it reduces the training time for each epoch.

Main characteristics of AlexNet implementation can be summarized in four aspects: (a) Data augmentation is carried out to reduce over-fitting. This data augmentation includes mirroring and cropping the images to increase the variation in the training data-set. The network uses an overlapped max-pooling layer after the first, second, and fifth CONV layers. Overlapped maxpool layers are simply maxpool layers with strides less than the window size. The 3x3 maxpool layer is used with a stride of 2 hence creating overlapped receptive fields. This overlapping improved the top-1 and top-5 errors by 0.4% and 0.3%, respectively. (b) Before AlexNet, the most commonly used activation functions were sigmoid and tanh. Due to the saturated nature of these functions, they suffer from the Vanishing Gradient (VG) problem and make it difficult for the network to train. AlexNet uses the ReLU activation function which doesn't suffer from the VG problem. The original paper (Krizhevsky et al. , 2012) showed that the network with ReLU achieved a 25% error rate about 6 times faster than the same network with tanh non-linearity. (c) Although ReLU helps with the vanishing gradient problem, due to its unbounded nature, the learned variables can become unnecessarily high. To prevent this, AlexNet introduced Local Response Normalization (LRN). The idea behind LRN is to carry out a normalization in a neighborhood of pixels amplifying the excited neuron while dampening the surrounding neurons at the same time. (d) AlexNet also addresses the over-fitting problem by using drop-out layers where a connection is dropped during training with a probability of p=0.5. Although this avoids the network from over-fitting by helping it escape from bad local minima, the number of iterations required for convergence is doubled.

## 2   Methodological testing site

### 2.1   Seismic monitoring of Lascar volcano

The Lascar volcano (23°22′ S, 67°44′ W; 5.592 m a.s.l.) is located in northern Chile,  270 km NE from Antofagasta and 70 km SE from San Pedro de Atacama, on the western border of the Altiplano-Puna 'plateau' (Figure 1). Lascar is considered the most active volcano in the Central Andean Volcanic Zone (de Silva and Francis , 1991). It is a compound elongated strato-volcano, comprised of two truncated western and eastern cones (Gardeweg et al. , 1998) that host five nested craters aligned ENE-WSW. The Lascar volcano has been seismically monitored by the CKELAR-VOLCANES group using a temporal network of 11 three-component stations (Shallow Posthole Seismometers, Model F72-2.0). These short-period 2 Hz seismometers were monitored continuously at 200 Hz from March to October 2018 in this first step of processing. Notably, only the Z channel was considered in building our database, the reason for this is that the spectrograms obtained in the different channels of a particular station are very similar, but in the case of the Z channel, the P phase is clearly identified for tectonic and volcano-tectonic events; the use of the other channels is reserved for future studies.

## 2.2 Lascar's database

Lascar's database corresponds to a catalogue of 6,145 seismic events, from which only 3,947 can be classified as volcanic events. The others, based on the distance to the hypocentres, are mainly tectonic events (not directly related to volcanic activity) recorded by Lascar's network during the period of observation. To guarantee the reliability of the database regarding volcanic activity, all observations were manually segmented, labelled, and checked from the continuous seismic record by CKELAR-VOLCANES experts. The processing routines consider the following 4 steps.

(a) Signal detection: The analysts detect the signal from the continuous seismic record using Seisan software (Havskov and Ottemöller , 2000), once detected, they proceed to write down the start and end times of the signal in a list (based on the duration times of the event for each station, Figure 2a).

(b) Preliminary classification: The analysts give, as appropriate, a preliminary label hybrid (HY), long period (LP), tectonic (TC), tremor (TR), volcano-tectonic (VT) to the event. Both, detection and preliminary classification, are based solely on visual observation of its raw waveforms (seismograms) from the different stations that recorded the event (Figure 2b).

(c) Classification: The analysts, using the duration time of each event and the Obspy package (Beyreuther et al., 2010), trim the signal, apply a linear detrend and a bandpass filter between 0.5-25 [Hz]. After that, they proceed to plot, one by one, the seismograms of the different stations, their amplitude spectra, and their spectrograms. Therefore, they decide by visual inspection of the frequency content and the seismogram of all stations that recorded the event, the respective class (Figure 2c).

(d) Signal segmentation: The analysts, using the list of absolute time of durations of each event, proceed to the segmentation of the signal to prepare an isolate corpus of seismograms as database (Figure 2d). The selection of the station for each event is decided by visual inspection of the frequency content and the seismogram of all stations that recorded the event, in relation to the level of noise in both the seismograms and the spectrograms (Figure 3).

Finally, the database used for the automatic classification experiment corresponds to isolated corpus of seismograms of five classes of events (Figure 4 and Figure 5): 213 events cataloged as (HY), 577 events as (LP), 471 events correspond to (TR), 2,686 events were recorded as (VT) and 2,198 events, not related to volcanic activity, were classified as (TC) (Tables 1 and 2).

## 3 Methods

We proposed a framework based on transfer learning, consisting of a previously trained deep convolutional neural network called AlexNet (Krizhevsky et al. , 2012). The main reasons for adopting this approach were, on one hand, to avoid the steps of an extensive search and selection of features, and, on the other hand, to deal with the limited number of labelled data. The data considered as input were spectrograms of the labelled dataset (waveform) from each class of seismic event. The line of processing consists of five steps: the first step describes the process applied in the time series (seismogram); the second step details obtaining spectrograms from the labelled seismograms; the third step consists of using the data augmentation technique to increase the number of data in the training and test datasets; the fourth step indicates how the prediction model is built; and, finally, the fifth step is performed to estimate the model's performance. In the following paragraphs, each step is explained in detail.

Step 1 (pre-processing): The observations were recorded with the same type of sensors, but at times the sample ratio should have been varied because of technical problems. Thus, many of the stations recorded at 200 Hz, but a few stations recorded at 500 Hz or 100 Hz. Considering this, we decided to apply 2 processing alternatives (a) resample the entire time series at 100 Hz, thereafter, we removed the mean and normalised the signals. Following this process, we applied a 10th order Butterworth bandpass filter between 1 and 10 Hz; (b) remove the mean and normalised, to apply the 10th order Butterworth bandpass filter

between 1 and 20 Hz and after then resample the time series to 100 [Hz]. The entire pre-processing was implemented using ObsPy (Beyreuther et al. , 2010).

Step 2 (spectrogram): The spectrograms were calculated by applying a Short-Time Fourier Transform, using the formula:

$$S[x(t)](n,k) = \left| \sum_{m=0}^{N-1} x(m) \cdot w(m-n) \cdot e^{i2\pi mk} \right|^2 \tag{1}$$

where $x(t)$ and $y(t)$ represent the seismic signal and short-time fourier transform sliding window, respectively. The sliding

window size was set to 1 s with a 95% overlap. The frequency interval used to calculate the spectrograms ranged from 1 to 10 Hz. All spectrograms were transformed to RGB images of $224 \times 224$ pixels.

Step 3 (data augmentation): Due to the imbalance produced on the labelled data, we decided to apply the data augmentation technique to generate a balanced number of different classes of seismic events (Table 2). Considering all the techniques for data augmentation, a time stretch was chosen. This transformation was implemented by rotations around the frequency axis

that were as random as possible, between 5-25% of the length of the signal, which were applied at the beginning, middle, or ending of the spectrogram as appropriate (Figure 6). The amount of data that was created for each class depended on the number of events in the most populated class; in this particular case the VT class was used as reference. The new signals generated were processed using the same procedure applied to the original signals (see Steps 1 and 2). The amount of synthetic data was created by considering all possible combinations of integers that represent a rotation angle between 5-25, that is, 21

possibilities. Subsequently, we apply these 21 rotation angles to the 3 possible rotation axes, corresponding to a total of 63 combinations of transformations to each event, without considering the repetition in the combination. Thus, each class has several synthetic data according to this relationship (63 times number of event per class). According with the transformations exposed above, the synthetic database was created using two criteria: (a) Considering the least populated class (HY) with 213 events, the number of events selected for each class must be equal to the maximum number of synthetic data that can be

generated in the (HY) class, it means 13,419 events, to obtain a balanced dataset. Thus, in the case of the (HY) class, all the synthetic data will be used, and for the others class, a random selection procedure, without repetition, will be implemented. Therefore, the number of events for this synthetic database comes to 67,095. (b) the amount of data that was created for each class depended on the number of events in the most populated class; in this case the (VT) class was used as reference (2,686 events). Thus, the number of events for each class must be equal to the 2,686 and the amount of synthetic data that must be

selected per each class depending on the number of events necessary to reach this quantity. Therefore, the total number of events for this database will be 13,430.

Step 4 (AlexNet): AlexNet is a deep convolutional neural network proposed by Krizhevsky et al. (2012) to classify the 1.2 million high-resolution images in the ImageNet LSVRC-2010 contest into 1,000 different classes. AlexNet was used in our study as a pre-trained deep convolutional neural network for spectrogram recognition, mainly because the spectrogram can be easily represented as an RGB image of $224 \times 224$ pixels.

Step 5 (model performance): This step was considered to evaluate the performance of the model and, thus, to validate the classification. This step was executed through a set of tests composed of signals that were not considered in the training step. We considered the following measures from the TorchMetrics in PyTorch:

i) Accuracy:

$$Accuracy = \frac{1}{N} \sum_{1}^{N} 1(y_i - \hat{y}_i) \tag{2}$$

where $y_i$ and $\hat{y}_i$ are the tensor of the target values and the tensor of the predictions, respectively.

ii) F1-score:

$$F1 - score = 2 \times \frac{Recall \times Precision}{Recall + Precision} \tag{3}$$

iii) Recall:

$$Recall = \frac{TP}{TP + FN} \tag{4}$$

where $TP$ and $FN$ represent the number of true positives and false negatives, respectively.

iv) Precision:

$$Precision = \frac{TP}{TP + FP} \tag{5}$$

where $TP$ and $FP$ represent the number of true positives and false positives, respectively.

v) Cohen's kappa:

$$\kappa = \frac{p_0 - p_e}{1 - p_e} \tag{6}$$

where $p_0$ is the empirical probability of agreement and $p_e$ is the expected agreement when both annotators assign labels randomly. Note that is estimated using a per-annotator empirical prior over the class labels.

## 4  Results

Due to the fact that our database is clearly imbalanced, with 3% (HY), 9% (LP), 36% (TC), 8% (TR), and 44% (VT) events (Table 2), we suspect that this natural behaviour most likely affect the automatic classification performance. Therefore, we

designed five experiments based on different dataset, to test the ability of the model to classify the data. The build of the dataset is explained in the following paragraphs.

(i) Corpus of real data (6,145 events): imbalanced database of the Lascar volcano was used, without the application of augmentation processes. Thus, the 80% of real data (4,916 events) were used to build the probabilistic model of classification and the other 20% (1,229 events) were used to test and validate this model (Table 3). This experiment permits us to evaluate the influence of imbalanced database in the performance of probabilistic model based on transfer learning (Alexnet).

(ii) Synthetic data corpus (67,095 events): where the synthetic data was created by data augmentation processes (for more details, see Step 3 in section 3). Once the synthetic database was created, 80% of it (53,676 events) was used to build the probabilistic classification model and the other 20% (13,419 events) was used to test and validate it (Table 3). This experiment allows us to assess the usefulness of our implementation of the data augmentation technique and whether, with synthetic data, transfer learning (Alexnet) was able to build a probabilistic model with good performance for automatic event classification.

(iii) Combined 1 (59,821 events): the experiment consisted of using the previous probabilistic model (ii) training with 53,676 events and testing it with all events of the real database of 6,145 events (Table 3). In this case, the performance of the probabilistic model built with synthetic data is evaluated on real data and, therefore, it will allow validating the efficiency of the data augmentation technique.

(iv) Combined 2 (13,430 events): this experiment used real and synthetic data to build and test the model. Thus, in the case of training we use 10,744 events, and for testing we use 2,686 events (Table 3). This approach will allow evaluating whether or not the amount of synthetic data created by data augmentation plays a key role in the performance of the probabilistic model to automatically classify events.

(v) Real database subset (1,213 events): the experiment consists of selecting a subset of real data, where the amount of data in each class depends on the number of events in the least populated class (HY) with 213 events. Thus, considering the class (HY) as a reference, we estimate that the number of events for the other classes will be 250 events; where 970 events correspond to the training set of the model and the other 243 events correspond to the testing set (Table 3). This experiment allows to evaluate the performance of the probabilistic model built with a minimum amount of data that provides an almost balanced database, without considering the generation of synthetics data by a data augmentation process.

Tables 3 and 4 present the statistics and metrics of the different experiments. It is evident that in the case of the unbalanced database, in experiment (i), the performance of the experiment was inferior (56.2% accuracy); however, when the data augmentation process was applied, in experiments (ii–iv), the results reached a particularly good percentage of the metric parameters (98.9%, 98.7%, 90.6% accuracy, respectively). In the case of the balanced real data subset experiment (v), the performance of the experiments was higher than experiment (i) 65.2% in accuracy, but notably lower than the experiments executed using the database with balanced classes (ii-iv). We want to highlight the results of the experiment, where the real data were used exclusively in the test (without being included in the model building). In this case, the metrics showed the second highest ranking, preceding only the experiment that exclusively included synthetic data.

In analysing the complete training and validation phase, through the loss and accuracy epoch, the process was less effective when the (i) database was used (Figures 7a-7b), whereas the best performance was achieved by the process using the (ii)

database (Figures 7c-7d). In the case of the database balanced with data augmentation and real data (iv) database, this showed a good performance (Figures 8a-8b). Conversely, the performance for the experiment with the database balanced only using real data (v), the results showed that despite having a balanced database, these are below the results obtained using data augmentation (Figures 8c-8d).

To evaluate the accuracy of the classification process for each experiment, we computed the confusion matrix. In the case of experiment (i), we noticed that classes with less data were negatively affected. In this case, a high percentage of HY events were confused by TC events and, thus, the TR events were also mistaken for TC events (Figures 9a-9b). Conversely, the classification process of experiment (ii) performed well, and no confusion in the class recognition was found (Figure 9c-9d). For experiment (ii), the performance was high, considering that the confusion of classes was practically minimal (Figures 10a-10b). Experiment (iv) also presented a high score, and the confusion matrix indicated a small problem in the recognition of TC and VT classes, in which some are confused by TR and TC, respectively (Figures 10c-10d). In the case of experiment (v) the main recognition problems are in the HY and TC class, in which some are confused with TR and LP (Figures 11a-11b).

To test whether the probabilistic model, built with both real and augmented data, is stable under the variability of the input data, a k-fold cross-validation procedure was implemented (Table 5). The results showed that the metrics for the different classes, as in the previous experiments, were high. There was only one class (TC) affected (with more variability), but good scores were always maintained, which in the worst case was over 67%.

In the case of the experiments that used a frequency band of 1-20 [Hz] (see Step 1, "case b" in section 3), the results showed a relative improvement compared to "case a" (frequency band 1- 10 [Hz]), but this is not comparable to the improvement achieved by applying the data augmentation process. The best improvement was achieved in experiment (i), where there is 6% difference in accuracy (Table 6).

## 5 Discussion

The analyses of this study clearly showed the impact of imbalance in the database and how the process of machine learning was conditioned to build probabilistic models of classifications for the different classes. The model building process was notably influenced by classes with more events, which in this case were TC and VT. Under these conditions, an over-fitted model was built in the training phase, which largely coincided with the recognition of these two classes, to the detriment of the less numerous ones.

Conversely, the process of data augmentation facilitated a balance in the data of the different classes, which directly impacted the performance of the probabilistic model, reaching optimal scores over different test datasets. This can be explained by the fact that data augmentation provides an efficient process for searching the features of each class during the training process.

We want to highlight the results of the experiment, where the real data were used exclusively in the test (without being included in the model building). In this case, the metrics showed the second highest ranking, preceding only the experiment that exclusively included synthetic data.

Although the frequency content was the main classification characteristic to differentiate the different volcanic event types, the choice of stretch as a data augmentation method along the time axis was a successful strategy that did not reduce effectiveness. The artificial production of data using stretches in frequency should provoke overlap (on the dimensional map) between the different classes of volcanic events when these are analysed using the spectrogram image.

The experiment (v), where we use a balanced real data subset, shows us that if we train with a balanced database, but the events per class are not enough, the results do not reach the values obtained with the experiments carried out with data augmentation as a mechanism to balance the classes. Thus, the probabilistic model built is not fully realiable, and there are more quantities of events confused with other classes.

Another important topic is the technique used to build the probabilistic model. Transfer learning with a pre-trained large neural network, such as AlexNet, facilitated model building in less time. Instead of manually selecting the best features, this task was performed by learning features from the pre-trained models of AlexNet, thereby dramatically saving time in the process. This point was verified when the model built using AlexNet was validated using only real data in experiment (iii), where high scores in the metrics of the experiment were achieved.

Thus, the use of transfer learning approach allowed the transference of the information about the features characteristics collected from the training dataset to the testing dataset, improving the efficiency of the process.

The limitations found in our implementation of Alexnet on spectrograms correspond mainly to the fact that although Alexnet has its own data augmentation process, it was not efficient enough to counteract the imbalance in the different classes studied. This shortcoming was solved by implementing our own data augmentation process.

Other limitation was that spectrogram must be scaled to a fixed input size of 224x224x3 pixels, due to the presence of fully connected layers in the convutional neural network. While this is not a big problem, it limits the graphical resolution of the image, although this can be advantageous since our main objective is to obtain a generalized probabilistic model.

A key issue for the proposed methodology was to generate a multi-station probabilistic model, developing a system trained with a multi-station subset and confronted with another independent multi-station subset. This approach permits us to reach a generalized set of characteristics that defined an optimal space for each class. Thus, we can avoid biases related to site effects (e.g., instruments installed in rocks, soil, etc.).

## 6 Conclusions

From the experiments implemented in this study, the following conclusions were drawn.

First, the usefulness of a multi-station framework to build a probabilistic model of the Lascar database allowed us to obtain a more generalisable model, thereby avoiding the bias associated with the choice of a particular station to retrieve the features. This fact led us to believe that we are very close to obtaining high-score results, with the AlexNet tool playing a key role in reaching the challenge of building a multi-volcano probabilistic model to classify the seismic events.

Second, data augmentation plays a key role as the main factor to improve the metrics of the experiments, thereby providing a built model validated by real data.

Last, and perhaps the most relevant, the proposal based on transfer learning provided an efficient feature retrieval process using learning features. The performance of this approach was clearly superior when compared with an exhaustive process of evaluation for a list of an hundred statistical features sent to the system, as it is a process of designed features. Our approach has a high impact when the time process matters, as is the case in early warning systems for volcanic activity, and provides a more generalisable model of prediction.

Following the same methodology used in the case of multi-station, we can expand, in a near future, our probabilistic models considering a network with different type of instruments, different setup of the instruments in a network, different temporality of the data analysed and, finally, reach a probabilistic model muti-volcano. The portability generated to apply this methodology will permit to work to different scales of operation of a network (from small temporal networks to world-wide volcano observatories). The acquisition of more generalisable models creates a good opportunity to develop a multi-volcano probabilistic model for volcanoes worldwide.

*Data availability.* The data are registered by DOI: 10.5281/zenodo.6001869. These can be founded in the link:

https://zenodo.org/record/6001870#.YkGXKffQ-Xl

*Author contributions.* Susana Layana, Gonzalo Yáñez y Pablo Salazar designed the experiments and carried them out. Franz Yupanqui, Claudio Meneses y Pablo Salazar developed the model code and performed the simulations. Pablo Salazar prepared the manuscript with contributions from all co-authors.

*Competing interests.* The author has declared that there are no competing interests.

*Acknowledgements.* This work has been funded by ANID/FONDECYT-INICIACIÓN/11190190, Antofagasta Regional Government, FIC-R project, code BIP N°30488832-0, Millennium Institute on Volcanic Risk Research-Ckelar Volcanoes ANID/MILENIO/ICN2021_038 and by Research Center for Integrated Disaster Risk Management (CIGIDEN), ANID/FONDAP/15110017. Susana Layana was funded by ANID-PCHA Doctorado Nacional 2016-21160276 fellowship and currently funded by Millennium Institute on Volcanic Risk Research-Ckelar Volcanoes ANID/MILENIO/ICN2021_038 through a postdoctoral fellowship. Franz Yupanqui is funded by Millennium Institute on Volcanic Risk Research-Ckelar Volcanoes ANID/MILENIO/ICN2021_038 and UCN doctoral fellowship. Our acknowledgment to CKELAR-VOLCANES team: Gabriel Ureta, Javier Valdés, Christian Ibaceta, Felipe Aguilera, Felipe Rojas, Diego Jaldín, Álvaro Vergara, Alfredo Esquivel, José Pablo Sepúlveda, Manuel Inostroza, Cristóbal González. To Greg Wait for his priceless course of training in volcanic signal.

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

415

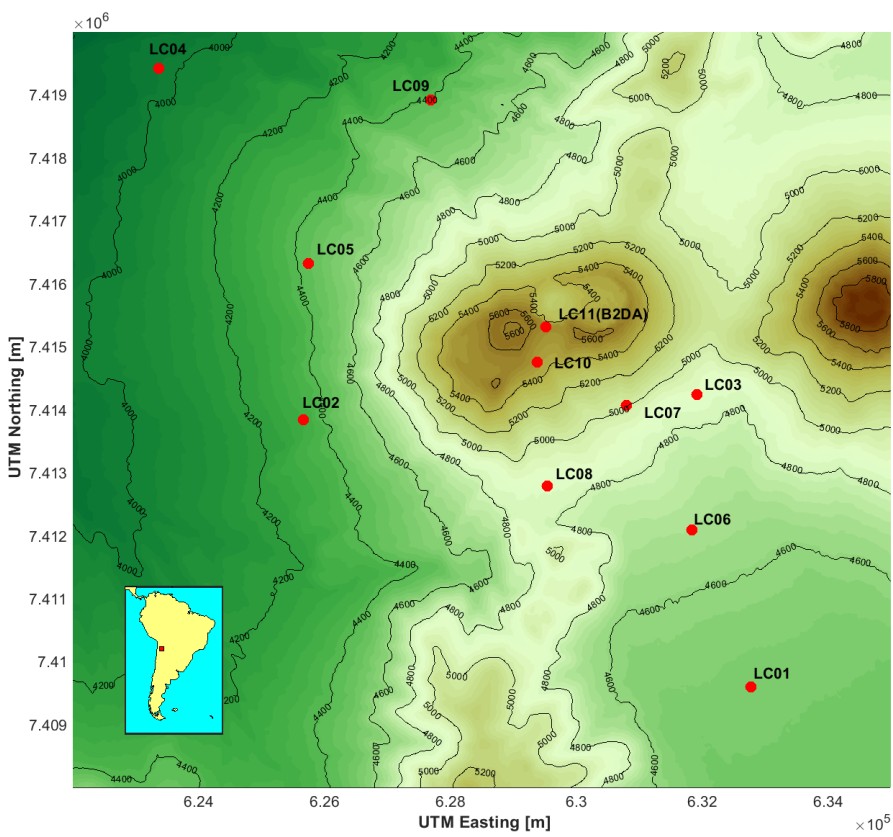

**Figure 1.** Location map of the Lascar volcano experiment. The red dots indicate the position of the short-period seismic stations and the black bold text represents the corresponding names. The LC10 and LC11 (B2DA) are located eastward, near the crater of the Lascar volcano. The DEM data is a product of ASTER Global Digital Elevation Model version 3 (ASTGTM v003), these can be downloaded directly by OPeNDAP link (https://lpdaac.usgs.gov/tools/opendap/). The procesing of the DEM was made using MATLAB ©.

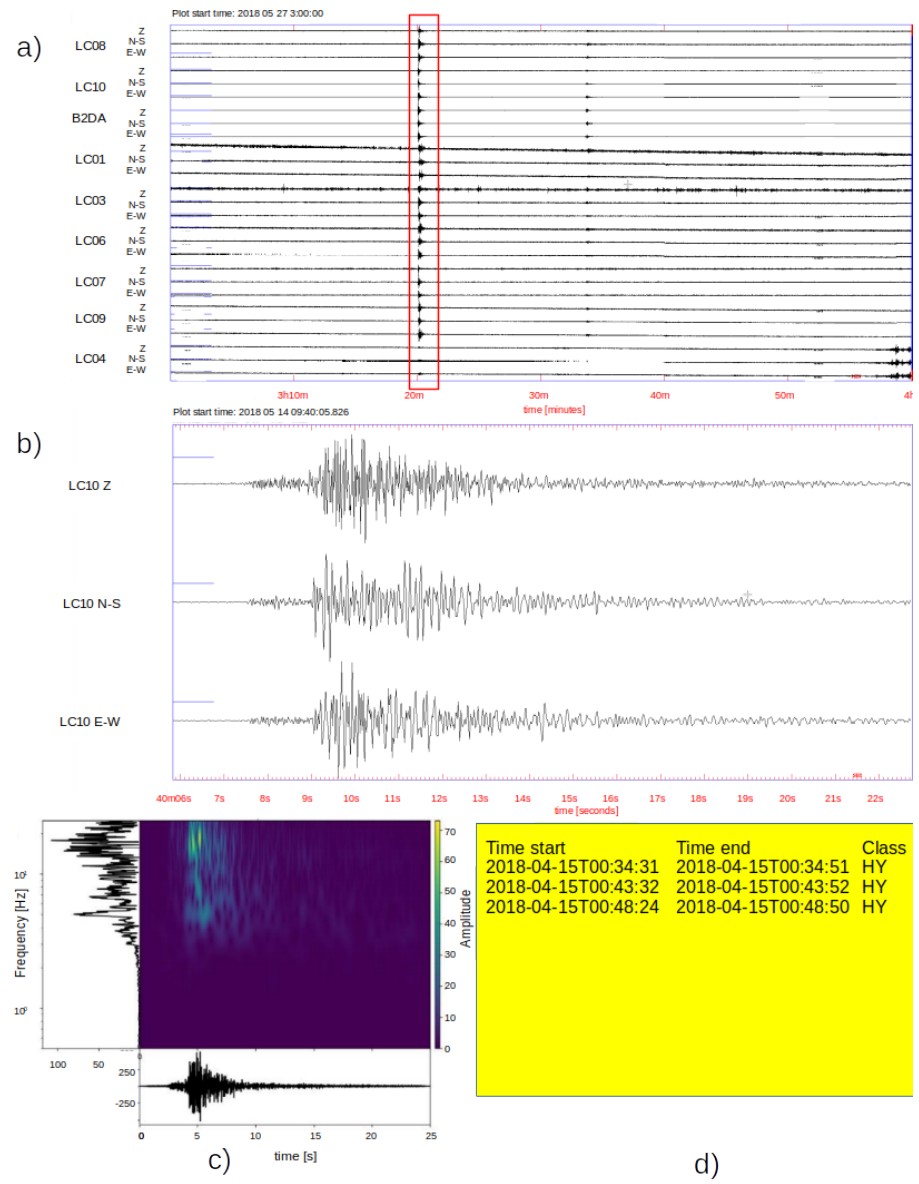

**Figure 2.** Building the Lascars database: a) signal detection; (b) preliminary classification; (c) classification; (d) signal segmentation.

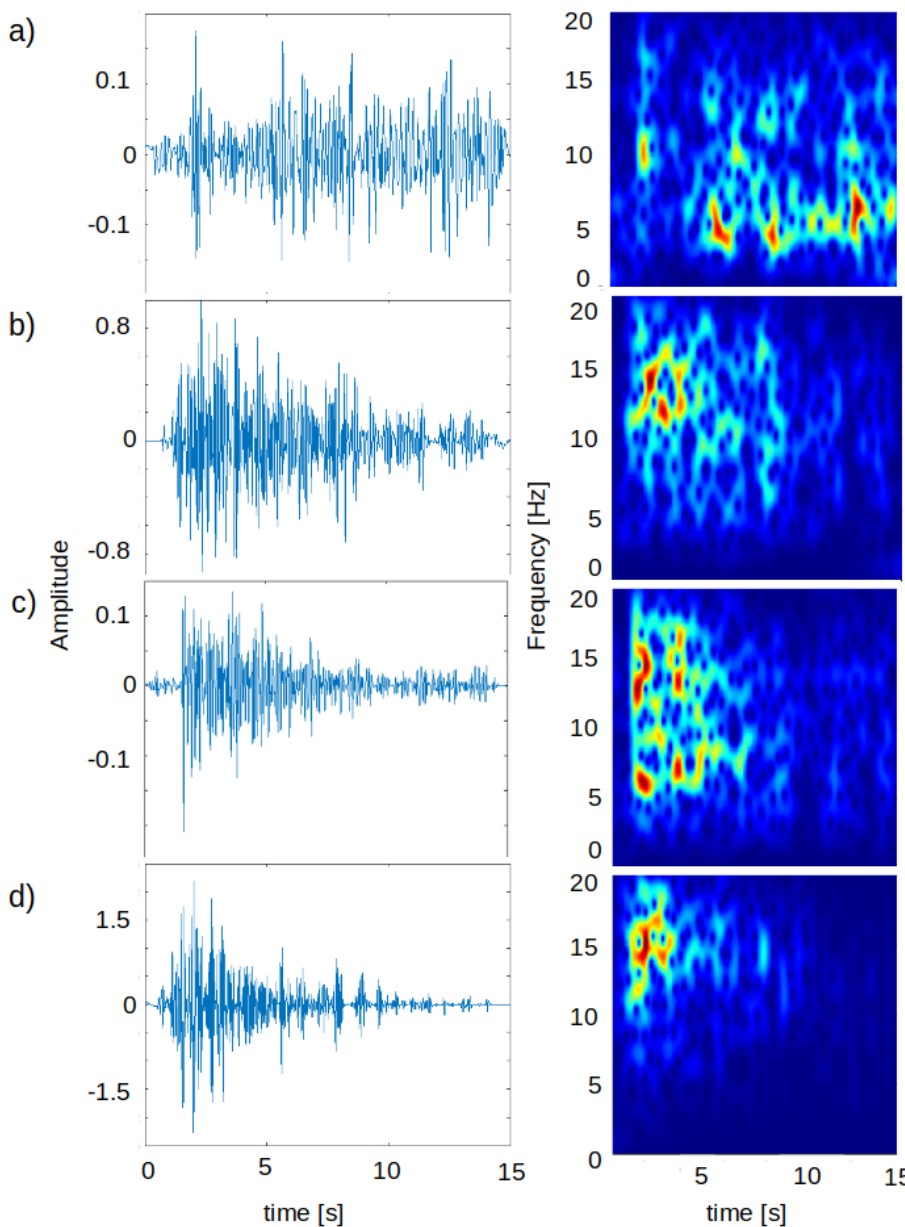

**Figure 3.** Station selection for each event: Example of station selection for the VT event that occurred on 2018-03-04T22:58:44, the event was recorded by the stations (a) LC01, (b) LC03, (c) LC04, (d) LC07. The spectrogram of the station LC07 was selected for its frequency content.

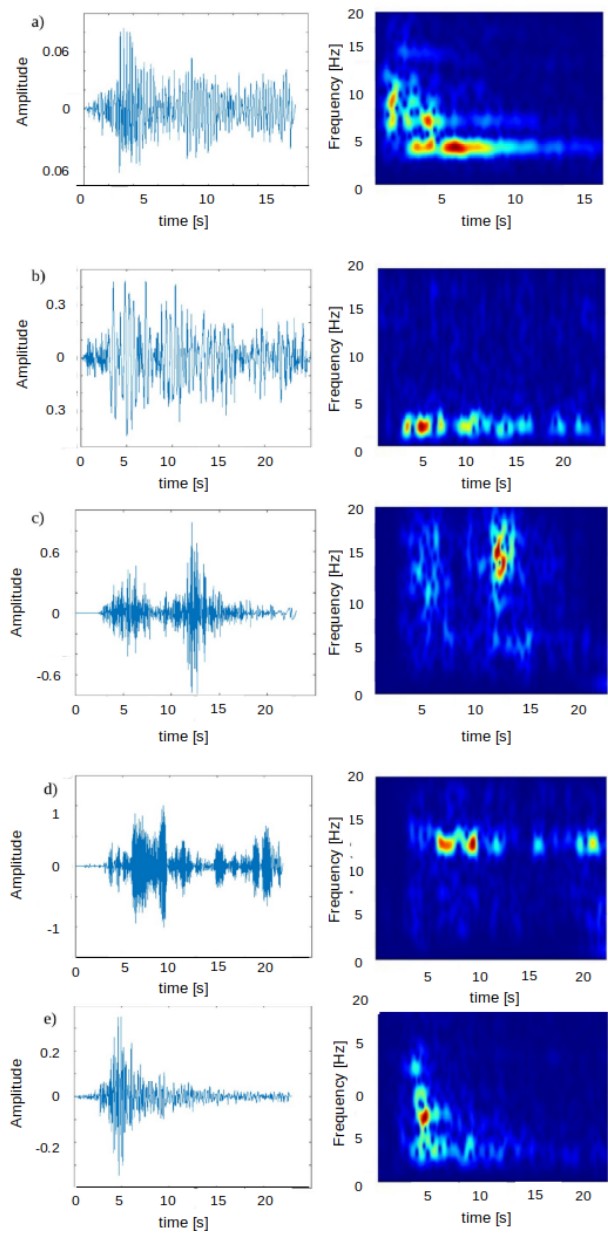

**Figure 4.** Examples of time series (left) and spectrograms (right) for the different classes in the Lascar database: a) hybrid events (HY), b) long period (LP), c) tectonic events (TC), d) tremors (TR), and e) volcano-tectonic (VT).

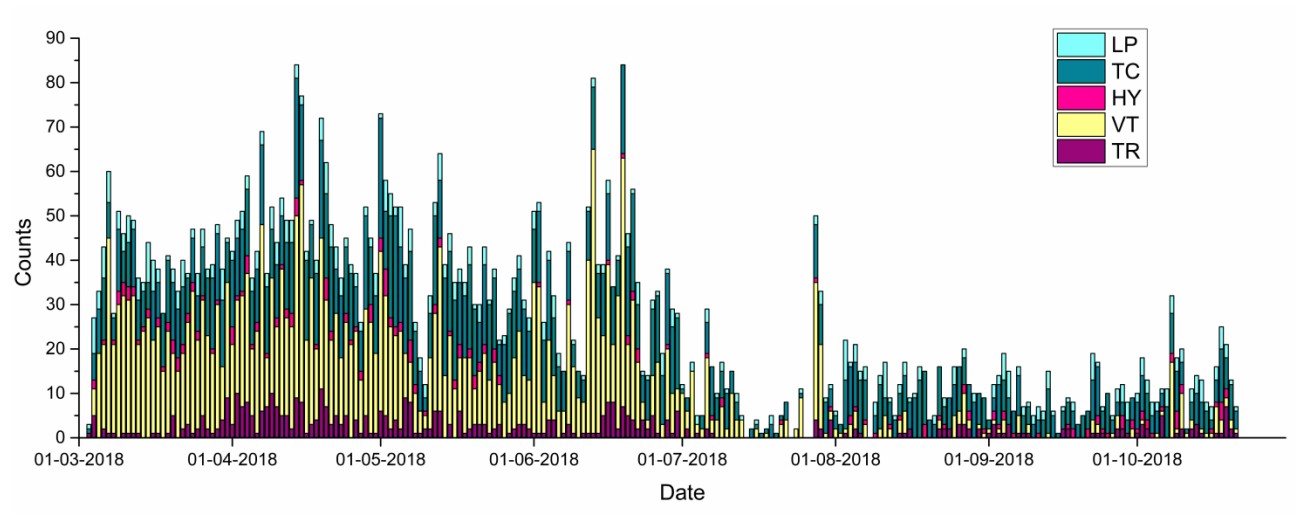

**Figure 5.** Temporal evolution in the production of events of the different classes for the Lascar volcano in the period since March to October, 2018.

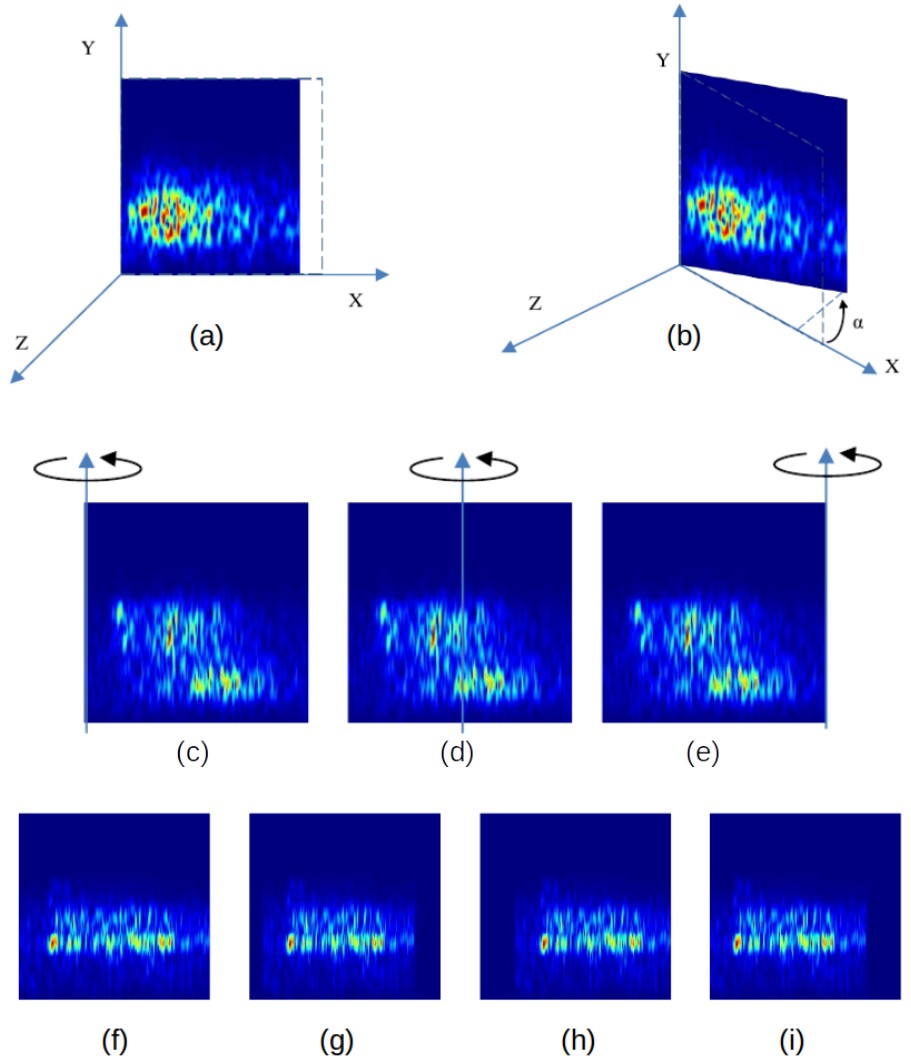

**Figure 6.** Transformation by rotations: the time stretching (a) is produced when we change the angle of rotation $\alpha$ around the frequency axis (b); examples of the hree possibilities of rotations, around a left (c), central (d), and right axis (e), respectively; examples of how the time stretching is produced by rotations, (f) original spectrogram, (g) rotation of 19% around the central axis, (h) rotation of 23% around a right axis, (i) rotation of 24% around a left axis.

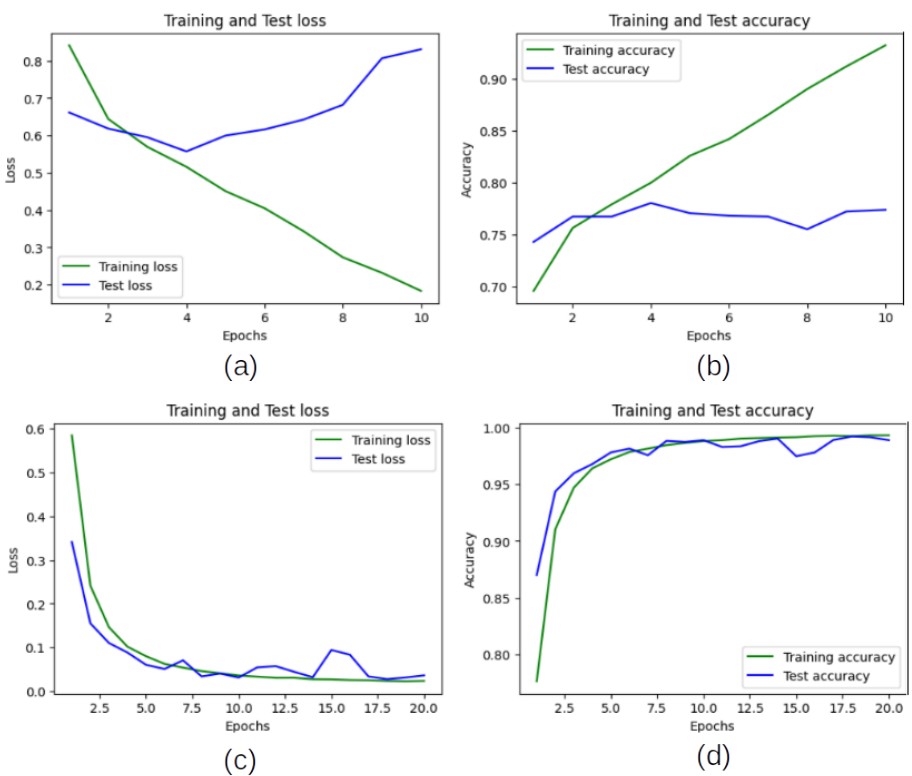

**Figure 7.** Training and validation performance scores of the transfer learning AlexNet for experiment: (a) Training and test loss for the experiment (i); (b) Training and test accuracy for the experiment (i); (c) Training and test loss for the experiment (ii), (d) Training and test accuracy for the experiment (ii).

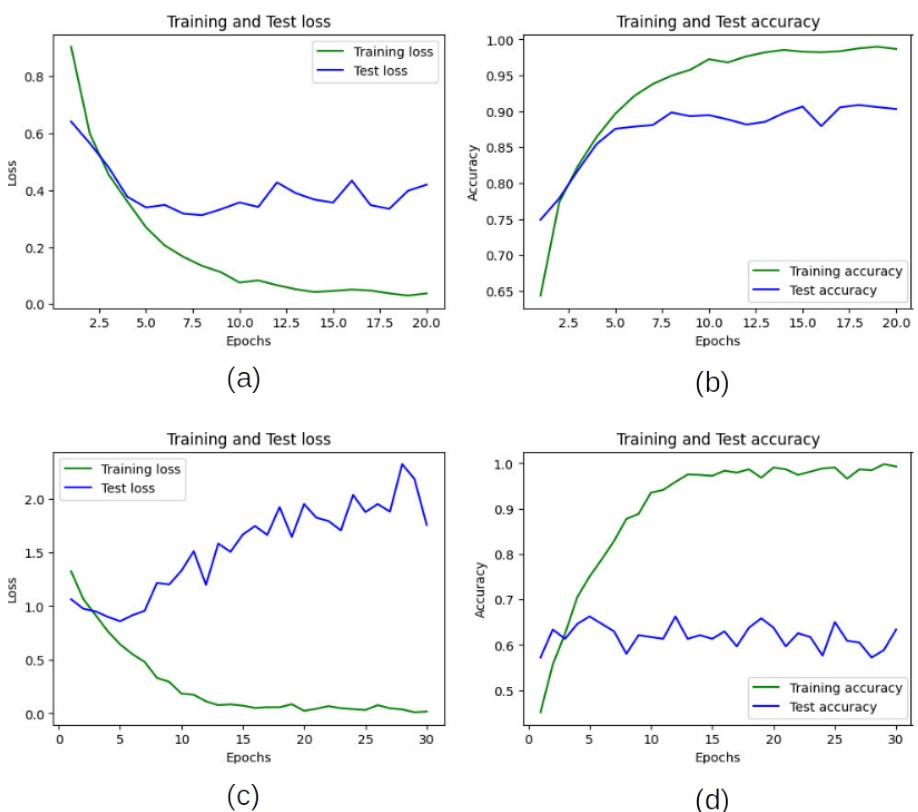

**Figure 8.** Training and validation performance scores of the transfer learning AlexNet for experiment: (a) Training and test loss for the experiment (iv), (b) Training and test accuracy for the experiment (iv); (c) Training and test loss for the experiment (v), (d) Training and test accuracy for the experiment (v)

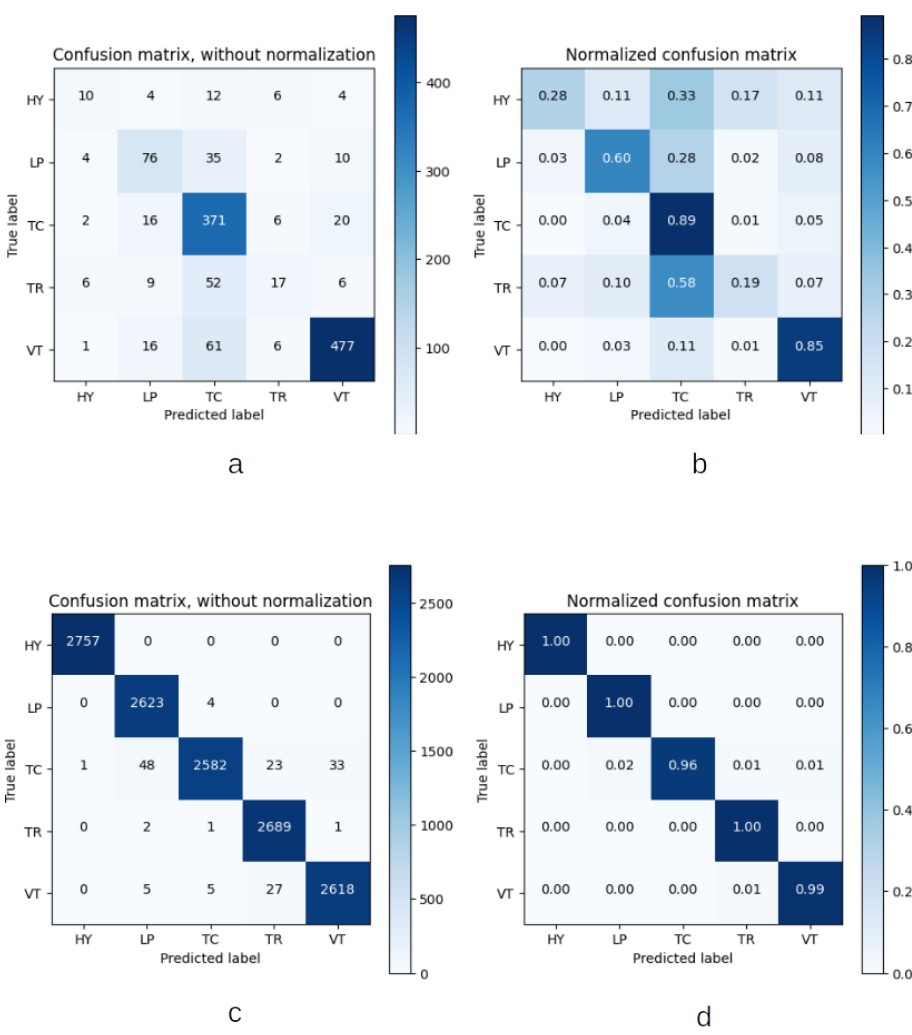

**Figure 9.** Confusion matrices: Without normalisation (a) and normalised (b), for experiment (i); Without normalisation (c) and normalised (d), for experiment (ii).

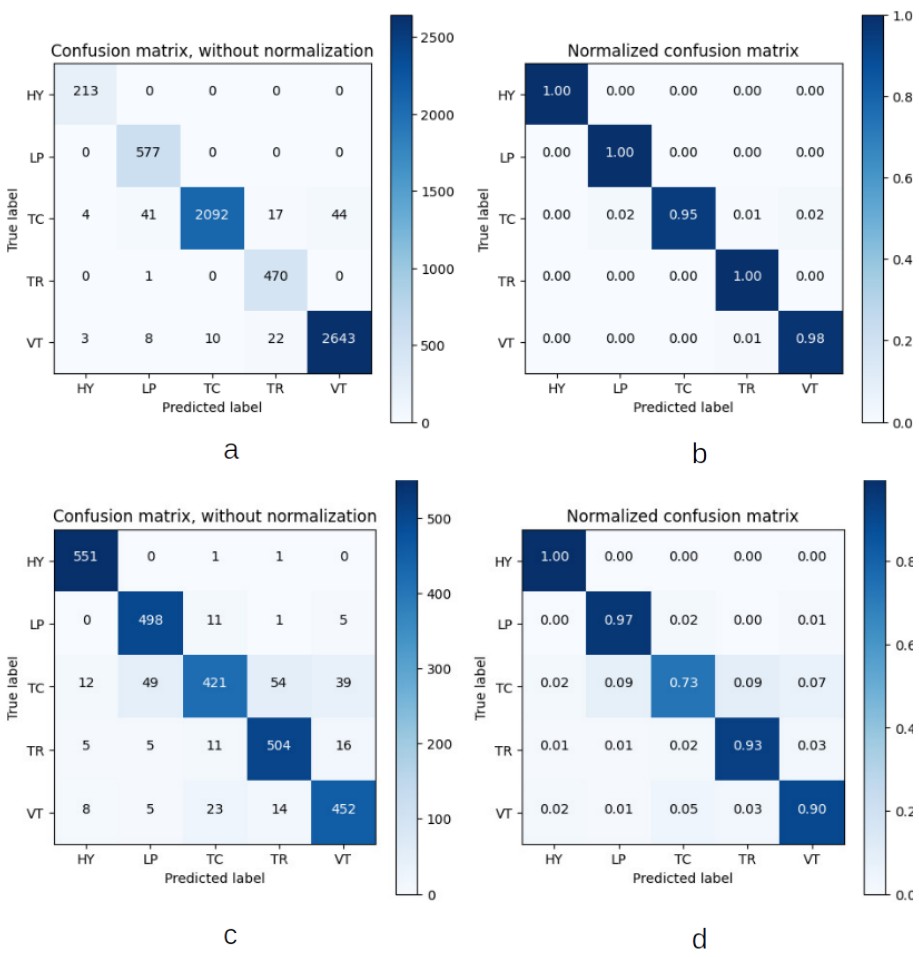

**Figure 10.** Confusion matrices: Without normalisation (a) and normalised (b), for experiment (iii); Without normalisation (c) and normalised (d), for experiment (iv).

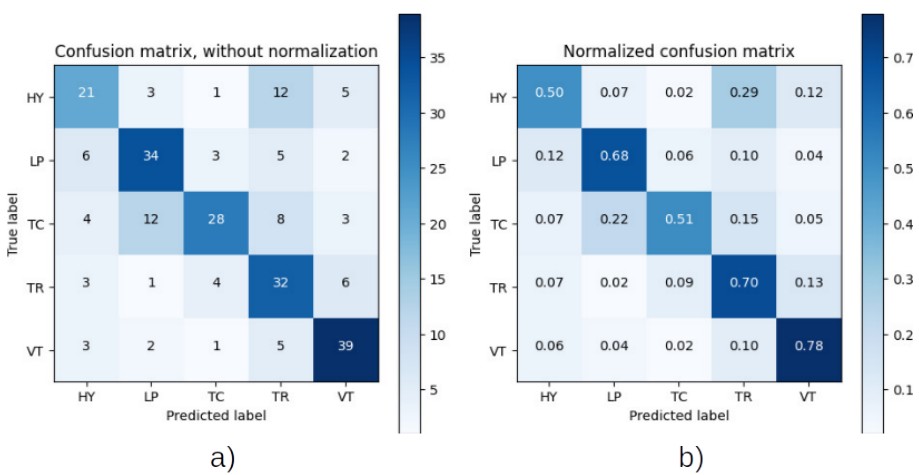

Figure 11. Confusion matrices: Without normalisation (a) and normalised (b), for experiment (v).

**Table 1.** Contribution of each station to Lascar's database.

| Classes/Stations | LC01 | LC02 | LC03 | LC04 | LC05 | LC06 | LC07 | LC08 | LC09 | LC10 | B2DA |
|---|---|---|---|---|---|---|---|---|---|---|---|
| HY | 0 | 0 | 0 | 7 | 45 | 4 | 0 | 3 | 12 | 136 | 6 |
| LP | 11 | 0 | 4 | 2 | 142 | 28 | 5 | 60 | 23 | 195 | 107 |
| TC | 9 | 0 | 11 | 97 | 284 | 36 | 157 | 127 | 144 | 1171 | 162 |
| TR | 1 | 0 | 5 | 29 | 21 | 4 | 61 | 13 | 35 | 292 | 10 |
| VT | 2 | 0 | 18 | 52 | 84 | 6 | 206 | 14 | 41 | 2249 | 14 |

This experiment considered only the Z channels (vertical).

**Table 2.** Statistics of the seismic records, based on the majority of the records.

| Classes/Events | Quantity % | Quantity N° | Maximum [s] | Minimum [s] | Mean [s] | Median [s] | Mode [s] | Standard deviation [s] |
|---|---|---|---|---|---|---|---|---|
| HY | 3 | 213 | 265 | 10 | 57.2 | 59 | 40 | 35.3 |
| LP | 9 | 577 | 600 | 9 | 82.3 | 67 | 25 | 63.5 |
| TC | 36 | 2198 | 670 | 9 | 78.5 | 65 | 20 | 56.2 |
| TR | 8 | 471 | 216 | 9 | 46.2 | 31 | 20 | 33.8 |
| VT | 44 | 2686 | 189 | 5 | 19.6 | 17 | 10 | 12.3 |
| Total | 100 | 6145 | 670 | 5 | 49.9 | 28 | 20 | 50 |

**Table 3.** Statistics related to the transfer learning experiments using AlexNet for hybrid events (HY), long period (LP), tectonic events (TC), tremors (TR), and volcano-tectonic (VT) classes.

| Corpus Experiments | Data Augmentation | Balanced data | Epoch N° | Total data 100% N° | Train data 80% N° | Test data 20% N° | Time to build the model [minutes] |
|---|---|---|---|---|---|---|---|
| (i) | No | No | 10 | 6145 | 4916 | 1229 | 4.6 |
| (ii) | Rotation 5–25% | Yes | 20 | 67095 | 53676 | 13419 | 164.9 |
| (iii) | Rotation 5–25% | Yes | – | – | – | 6145 | 0.3 |
| (iv) | Rotation 5–25% | Yes | 20 | 13430 | 10744 | 2686 | 10.6 |
| (v) | No | Yes | 25 | 1213 | 970 | 243 | 2.2 |

**Table 4.** Performance of the transfer learning experiments to 1-10 [Hz].

| Experiments/Metrics | Accuracy | F1 | Recall | Precision | Cohen's kappa | Class Accuracy | | | | |
|---|---|---|---|---|---|---|---|---|---|---|
| | | | | | | HY | LP | TC | TR | VT |
| (i) | 56.2 | 57.8 | 56.2 | 62.9 | 65.2 | 27.8 | 59.8 | 89.4 | 18.9 | 85 |
| (ii) | 98.9 | 98.9 | 98.9 | 98.9 | 98.6 | 100 | 99.8 | 96.1 | 99.9 | 98.6 |
| (iii) | 98.7 | 97.2 | 98.7 | 95.8 | 96.4 | 100 | 100 | 95.2 | 99.8 | 98.4 |
| (iv) | 90.6 | 90.2 | 90.6 | 90.3 | 87.9 | 99.6 | 96.7 | 73.2 | 93.2 | 90.0 |
| (v) | 65.2 | 65.1 | 65.2 | 65.5 | 56.3 | 66.7 | 66.0 | 58.2 | 67.4 | 68.0 |

**Table 5.** Performance of the transfer learning experiments applying k-fold validation.

| k-fold/Metrics | Accuracy | F1 | Recall | Precision | Cohen's kappa | Class Accuracy | | | | |
|---|---|---|---|---|---|---|---|---|---|---|
| | | | | | | HY | LP | TC | TR | VT |
| 1 | 89.2 | 89.0 | 89.2 | 89.2 | 86.7 | 99.3 | 95.2 | 75.3 | 84.8 | 91.3 |
| 2 | 90.1 | 90.1 | 90.1 | 90.4 | 87.5 | 90.1 | 99.6 | 89.9 | 84.7 | 86.6 |
| 3 | 89.3 | 89.0 | 89.3 | 89.1 | 86.6 | 89.3 | 98.3 | 97.3 | 71.8 | 90.7 |
| 4 | 89.8 | 89.5 | 89.8 | 90.1 | 87.2 | 89.8 | 97.3 | 98.8 | 67.8 | 90.6 |
| 5 | 87.9 | 87.7 | 87.9 | 87.8 | 84.6 | 87.9 | 93.1 | 87.8 | 77.7 | 89.1 |
| 6 | 88.6 | 88.4 | 88.6 | 88.7 | 85.6 | 88.6 | 99.6 | 91.3 | 81.5 | 77.9 |
| 7 | 87.6 | 87.7 | 87.6 | 88.1 | 84.8 | 87.6 | 96.6 | 87.5 | 82.7 | 80.6 |
| 8 | 89.8 | 89.6 | 89.8 | 89.7 | 87.1 | 89.8 | 98.6 | 91.8 | 75.2 | 89.9 |
| 9 | 88.7 | 88.7 | 88.7 | 89.0 | 85.9 | 88.7 | 98.9 | 89.6 | 78.5 | 83.5 |
| 10 | 90.4 | 90.2 | 90.4 | 90.2 | 87.4 | 90.4 | 96.0 | 91.2 | 79.9 | 94.9 |

**Table 6.** Performance of the transfer learning experiments for the experiment to 1-20 [Hz].

| Experiments/Metrics | Accuracy | F1 | Recall | Precision | Cohen's kappa | Class Accuracy | | | | |
|---|---|---|---|---|---|---|---|---|---|---|
| | | | | | | HY | LP | TC | TR | VT |
| (i) | 62.8 | 63.9 | 62.8 | 66.7 | 67.6 | 38.9 | 74.8 | 73.5 | 33.3 | 93.4 |
| (ii) | 99.4 | 99.4 | 99.4 | 99.4 | 99.3 | 100 | 99.8 | 98.2 | 100 | 99.2 |
| (iii) | 99.5 | 98.9 | 99.5 | 98.3 | 98.5 | 100 | 100 | 98 | 100 | 99.3 |
| (iv) | 91.5 | 91.4 | 91.5 | 91.4 | 89.2 | 98.4 | 92.4 | 83.5 | 93.3 | 89.6 |
| (v) | 64.2 | 64.1 | 64.2 | 66.2 | 55.2 | 50 | 68 | 49.1 | 73.9 | 80 |