# Peer review of "Multi-station automatic classification of seismic signatures from the Lascar volcano database"

_Natural Hazards and Earth System Sciences, 2022_

## Referee Comment (RC1)

The Authors applied a pre-trained convolutional network (AlexNet) to build a multi-station automatic classification system for volcanic seismic signals. They used data that were recorded by a temporal network installed in the Lascar volcano. The Authors designed and performed four experiments using different datasets with real data, synthetic data, and two different combinations of these (combined 1 and combined 2).

The authors obtain fairly good correct classification performances especially in some experiments (those in which they used a data augmentation technique).

I found the article interesting and worth of publishing in the Natural Hazards and Earth System Sciences journal. I have, however, some general and specific comments, which will require some revision of the article.

**General comments:**
My feeling is that with some changes in the pre-processing phase the performance can further improve. Furthermore, I find the use of spectrograms in image form rather original, however I do not think it is incorrect.

**Specific comments:**
The abstract must be improved because the reader does not understand which signals the Authors want to classify and what results they obtain in terms of method performance.

*Figures*
The labels of the figures must be larger (for example, Fig. 1, 2, 6, 7, 8, 9, 10, 11)
In Fig. 4 the letters a, b, c,

On lines 3, 48, 49 etc. "artificial data" should be replaced with "synthetic data".

Lines 69- 71 This paragraph does not seem clear to me. Please rephrase.

Lines 107 -111 In principle, in step 1 you should first remove the average, then filter (a 10th order filter seems a bit strong to me as a filter for a fairly high frequency band such as 1-10 Hz) and finally resample the series, however I think the ObsPy resampling routines apply an appropriate filter before resampling. Furthermore, looking at figure 2 you can see that tectonic events (TC) and tremor have a significant frequency content in the 10-20 Hz band (this is unusual for volcanic tremor which generally has lower frequencies, between 1 and 6 Hz), therefore, in the pre-processing step it would be more appropriate to filter in the 1 - 20 Hz frequency band.

Lines 117 -123 In step 3 to avoid the imbalance produced on the labeled data, the Authors could create a fifth dataset not applying the data augmentation technique, but selecting a subset of the data so that the amount of data that is selected for each class depends on the number of events in the less populated class; in particular the hybrid events HY class (213 events) that can be used as reference (the number of events for the other classes could perhaps be even slightly higher, for example 250 events for the other classes and 213 for hybrid events).

147 see lines 117 -123

Lines 152 -157 In this paragraph the authors should insert values of performance also in the text and not refer only to table 4.

Line 202 "This will improve the understanding and evaluation of the hazards and risks associated with the activity of volcanoes." I believe this last sentence is not necessary.

---

## Author Response (AR1)

Referee 1:

"The abstract must be improved because the reader does not understand which signals the Authors want to classify and what results they obtain in terms of method performance."

Answer: We add some comments in the lines 1-2; line 5, and lines 10-11.

"The labels of the figures must be larger (for example, Fig. 1, 2, 6, 7, 8, 9, 10, 11)"

Answer: We re-scaled the figures and labeled with an adequate font size.

"On lines 3, 48, 49 etc. "artificial data" should be replaced with "synthetic data"."

Answer: artificial data was replaced with synthetic data.

"Lines 107 -111 In principle, in step 1 you should first remove the average, then filter (a 10th order filter seems a bit strong to me as a filter for a fairly high frequency band such as 1-10 Hz) and finally resample the series, however I think the ObsPy resampling routines apply an appropriate filter before resampling. Furthermore, looking at figure 2 you can see that tectonic events (TC) and tremor have a significant frequency content in the 10-20 Hz band (this is unusual for volcanic tremor which generally has lower frequencies, between 1 and 6 Hz), therefore, in the pre-processing step it would be more appropriate to filter in the 1 - 20 Hz frequency band."

Answer: We added to the routine process your advice. Lines 165-168. We also added a Table 6 that summarized all the results and presents some comments about these in the lines 276-279.

"Lines 117 -123 In step 3 to avoid the imbalance produced on the labeled data, the Authors could create a fifth dataset not applying the data augmentation technique, but selecting a subset of the data so that the amount of data that is selected for each class depends on the number of events in the less populated class; in particular the hybrid events HY class (213 events) that can be used as reference (the number of events for the other classes could perhaps be even slightly higher, for example 250 events for the other classes and 213 for hybrid events)."

Answer: This new experiment was created and discuss in the text, lines 243-248; lines 252-254; lines 261-262; lines 269-270; lines 296-299.

"147 see lines 117 -123"

Answer: The new experiment was added to the Results, lines 243-248.

"Lines 152 -157 In this paragraph the authors should insert values of performance also in the text and not refer only to table 4."

Answer: The values were added to the paragraph line 250; lines 252-254.

"Line 202 "This will improve the understanding and evaluation of the hazards and risks

associated with the activity of volcanoes." I believe this last sentence is not necessary."

Answer: This sentence was removed from the text, lines 335-336.

Referee 2
"First, I would more extensively describe and discuss AlexNet discussing (a) its implementation, (b) target, and (c) recognized advantages and limitations. These should then be discussed in the light of the analysis of spectrograms, which is here proposed."

Answer: We added some paragraph in order to answer these comments: (a) implementation, (b) target and (c) recognized advantages and limitations. All these aspects are included in the lines 83-113. In the case of discussion, this aspect was included in the lines 305-312.

"Next, authors should better introduce the classification and elaborate on the different classes of signals. This classification is, I believe, inherited from previous manual catalog, where volcano-seismologists introduced the different classes on the base of visual observation of a large amount of data. How is the classification done in the original catalog? What are the typical features? Can one show a compact example of different waveform types and corresponding spectrograms?"

Answer: This information was added in the lines 131-152. Also the Figures 2 and 3 were added in order to explain the steps.

"Adding some more information on the typical spatial and temporal evolution of these type of events, and the different amount of signals in the original catalog, would also make the paper more interesting and easy to read."

Answer: We included a new figure, Figure 5 to show the temporal evolution and the different amount of signals in the catalog. We do not included spatial distribution, mainly because is not the focus of the article and we also have some location for VT events.

"From the technical point of testing the performance of the approach, it would be good to make explicit e.g. the advantage to add merged data-synthetic datasets. Here, I think the first part of the result section could be extended, accompanying the description of each processing with the aim of such test(s)."

Answer: We included some paragraphs in order to give more detail about this in the lines 224-248, lines 181-194.

"Authors use only Z component seismograms, and mention 3-component data may be used in future (in a future work). This is acceptable, but the authors should somehow justify this choice."

Answer: We included the lines 123-124 to explain this point.

"I am still confused about how the classification performed at different stations is joined into a single catalog. What, e.g. if differently detected and/or classified at different stations?"

Answer: This aspect was included in the lines 143-145. Also a new Figure was added Figure 3 in order to explain this.

"Results and discussion sections are too compact and not well separated: some of the text in the result section seems to be more adequate for the discussion."

Answer: Both, Results and Discussion was re-write in order to give more information and cover in the discussion more aspects.

"A key issue for the proposed approach, and the interest it may have for a broader volcano-seismology audience, seems to be portability of this method, e.g. to other volcanoes, other time spans, or other monitoring setup. This can only be partially discussed here, as only one volcano monitoring is considered. However, one can still play with different observations. Can we train with one station and process another one?"

Answer: This aspect was discuss in the lines 313-316.

"Figure 1 could be more informative, plotting the regions where different type of seismicity was found in the original catalog and to inform the reader where does seismicity take place. Station symbols are too small, as well as labels. The figure size is too small"

Answer: The Figure 1 was modified, the sesimicity was not included because is not the focus of our article.

"I would also add a figure showing the time evolution of detections of different classes over time for the study period; such figure could help to summarize these results."

Answer: The Figure 5 was included.

"Figure 2 is important and useful, but somewhere hard to read. Labels should be larger, to appreciate the (common) duration of the signal."

Answer: The former Figure 2, now Figure 4, was modified in the labels associated to the signal.

"Figs. 3-5 can be possibly be merged into a single figure with different panels. Here, again, labels are too small."

Answer: The Figures were modified.

"Same for Figs. 6 and 7, join in a single figure"

Answer: The Figures were modified.

"And same for confusion matrices (Figs. 8-11), they can be subpanels of a single figure, with no loss of information."

Answer: The Figures were modified.

---

## Author Response (AR2)

Comments of reviewer #1

List of comments:

Text:

L. 94-95. 'The number of classes … consists of 1000 classes'. Please, rephrase the sentence

Resp.: The phrase ha been changed.

L. 100 Data → data

Resp.: The word has been written with lowercase.

L. 132 'Signal detection' (with capital letter and, so on, Preliminary classification, Classification, Signal segmentation)

Resp.: The words have been changed to uppercase.

L. 185: avoid italic here

Resp.: The italic has been changed to normal text.

L. 226-248: I suggest also here to use capital letter at the beginning of each bullet point (i.e. (i) Corpus, (ii) Synthetic, and so on)

Resp.: The words have been changed to uppercase.

Figures:

Fig. 1. Axes labels and values are still too small. Perhaps increase dpi for a better quality of the figure (it is a bit blurred in my pdf).

Resp.: The labels of Figure 1 has been changed to bigger than older size.

Same problem in Fig. 2:
a) panels a and b include some text which I cannot even read
b) the horizontal scales of the same panels are not readable, and there is no label (i.e. 'time (s)'?).
c) Labels of panel c are too small
d) Panel d can be probably dropped

Resp.: All the topics have been solved doing the respective changes.

Figure 5 is missing in my pdf, as well as its label

Resp.: We have optimized the size of image in order than it do not has problem when is charged in the pdf file.